# Chiral and flat-band magnetic quasiparticles in ferromagnetic and metallic kagome layers

S. X. M. Riberolles [1], Tyler J. Slade[1], Tianxiong Han [1,2], Bing Li [1,2], D. L. Abernathy [3], P. C. Canfield[1,2], B. G. Ueland [1], P. P. Orth[1,2], Liqin Ke [1] & R. J. McQueeney[1,2] ✉

Magnetic kagome metals are a promising platform to develop unique quantum transport and optical phenomena caused by the interplay between topological electronic bands, strong correlations, and magnetic order. This interplay may result in exotic quasiparticles that describe the coupled electronic and spin excitations on the frustrated kagome lattice. Here, we observe novel elementary magnetic excitations within the ferromagnetic Mn kagome layers in $TbMn_6Sn_6$ using inelastic neutron scattering. We observe sharp, collective acoustic magnons and identify flat-band magnons that are localized to a hexagonal plaquette due to the special geometry of the kagome layer. Surprisingly, we observe another type of elementary magnetic excitation; a chiral magnetic quasiparticle that is also localized on a hexagonal plaquette. The short lifetime of localized flat-band and chiral quasiparticles suggest that they are hybrid excitations that decay into electronic states.

In condensed matter physics, the concept of a quasiparticle provides a powerful simplification that enables an intuitive understanding of interacting systems. Quasiparticles are associated with a material's elementary excitations and can be assigned charge, mass, spin, and other properties that are used to describe actual fundamental particles, such as a photons or an electrons. In topological materials, low-energy electronic excitations provide unique and testable manifestations of Dirac, Weyl, Majorana, and axion quasiparticles which have a long history in the development of particle physics and cosmology[1–3]. Magnetic materials are particularly rich systems for the emergence of novel quasiparticles since the breaking of time-reversal symmetry may result in exotic, correlated magnetic ground states. While in simple ferromagnets (FMs), magnon quasiparticles are massive spin-1 bosons that effectively describe the collective precession of the uniform magnetization, quantum antiferromagnets (AFs) can exhibit emergent spinless Higgs quasiparticles (amplitude modes), fractionalized charge neutral spin-1/2 spinons, or magnetic monopole excitations[4–6].

The combination of strong interactions with geometric frustration is fertile ground for the emergence of quasiparticles comprised of electronic, orbital, and magnetic degrees-of-freedom. Frustrated kagome lattices exhibit nearest-neighbor hopping that guarantees flat electronic bands, Dirac band crossings, and other band touchings. These features are shared by any quasiparticle moving within the kagome layer, independently of whether they carry charge or are neutral. The flat electronic bands are susceptible to a variety of instabilities driven by electronic correlations and band filling. Recent discoveries of superconductivity[7], charge ordering[8], and orbital loop currents[9,10] in $AV_3Sb_5$, itinerant ferromagnetism (FM) with large (topological) anomalous Hall response in $Co_3Sn_2S_2$[11] have elevated interest in kagome metals as an adaptable system to study the interplay of topology, superconductivity, magnetism, and other charge instabilities[9].

In FM kagome metals, the coupling of magnetic order and topological electrons leads to the emergence of massive Dirac and chiral Weyl fermionic quasiparticles in the charge sector. In the magnetic sector, the magnetic moments on a kagome lattice can also host spin-1 acoustic, optical, and flat-band magnon quasiparticles with non-trivial topology (eg. Dirac magnons) that are similar to the electronic counterpart[12–14]. The itinerant character of the magnetism opens key questions about whether new quasiparticles may arise from a hybridization between the charge and the magnetic sector. One exotic possibility that we find to be consistent with our experimental

[1]Ames National Laboratory, Ames, IA 50011, USA. [2]Department of Physics and Astronomy, Iowa State University, Ames, IA 50011, USA. [3]Oak Ridge National Laboratory, Oak Ridge, TN 37831, USA. ✉e-mail: mcqueeney@ameslab.gov

observations is the interaction of charged loop currents that induce correspondingly localized magnetic quasiparticles with a nonzero vector spin chirality.

In the following, we describe the discovery of magnetic chiral quasiparticles in FM kagome layers by measuring their elementary excitations using inelastic neutron scattering (INS). Recent INS studies of a variety of FM kagome metals, such as FeSn[15–18], Fe$_3$Sn$_2$[19–21], and TbMn$_6$Sn$_6$[22,23], reveal well-defined and collective acoustic magnon modes at low energies. However, the higher-energy optical and flat band magnon modes and their associated topological features were obscured by heavy damping[22,24–27]. Here, we perform experiments on much larger sample volumes and with increased incident neutron energies that reveal two broad, high-energy excitations in TbMn$_6$Sn$_6$ that are better described as localized magnetic quasiparticles, rather than collective magnon modes. The first excitation consists of dynamical spin correlations around a hexagonal plaquette, corresponding to the expected kagome Wannier states associated with localized flat-band magnon quasiparticles[28], thereby providing the clearest experimental evidence for a magnonic flat band in a kagome metal. The

second excitation exhibits unexpected chiral spin correlations around a hexagonal plaquette which are truly anomalous and cannot be captured from simple magnetic models. The observed short-lifetime of chiral and flat-band magnetic quasiparticles is caused by decay into other quasiparticles, likely of electronic origin, and points towards a strong hybridization between charge and magnetic sectors.

## Results

$R$Mn$_6$Sn$_6$ materials (where $R$ is a rare-earth) comprise an interesting class of magnetic kagome metals[29–34]. The metallic Mn kagome layers have robust itinerant FM order that can be manipulated by interleaved FM $R$ triangular layers through $R$–Mn AF coupling which generates a 3D ferrimagnetic structure, as shown in Fig. 1a. In TbMn$_6$Sn$_6$, $R$–Mn coupling and the uniaxial magnetic anisotropy of the Tb ions forces Mn moments to orient perpendicular to the kagome layer at low temperatures[35], creating an ideal scenario for a Chern insulator[30].

As there are two Mn kagome layers in the unit cell, TbMn$_6$Sn$_6$ is predicted to have seven magnon branches. We label these branches as acoustic–even (AE), acoustic–odd (AO), optical–even (OE),

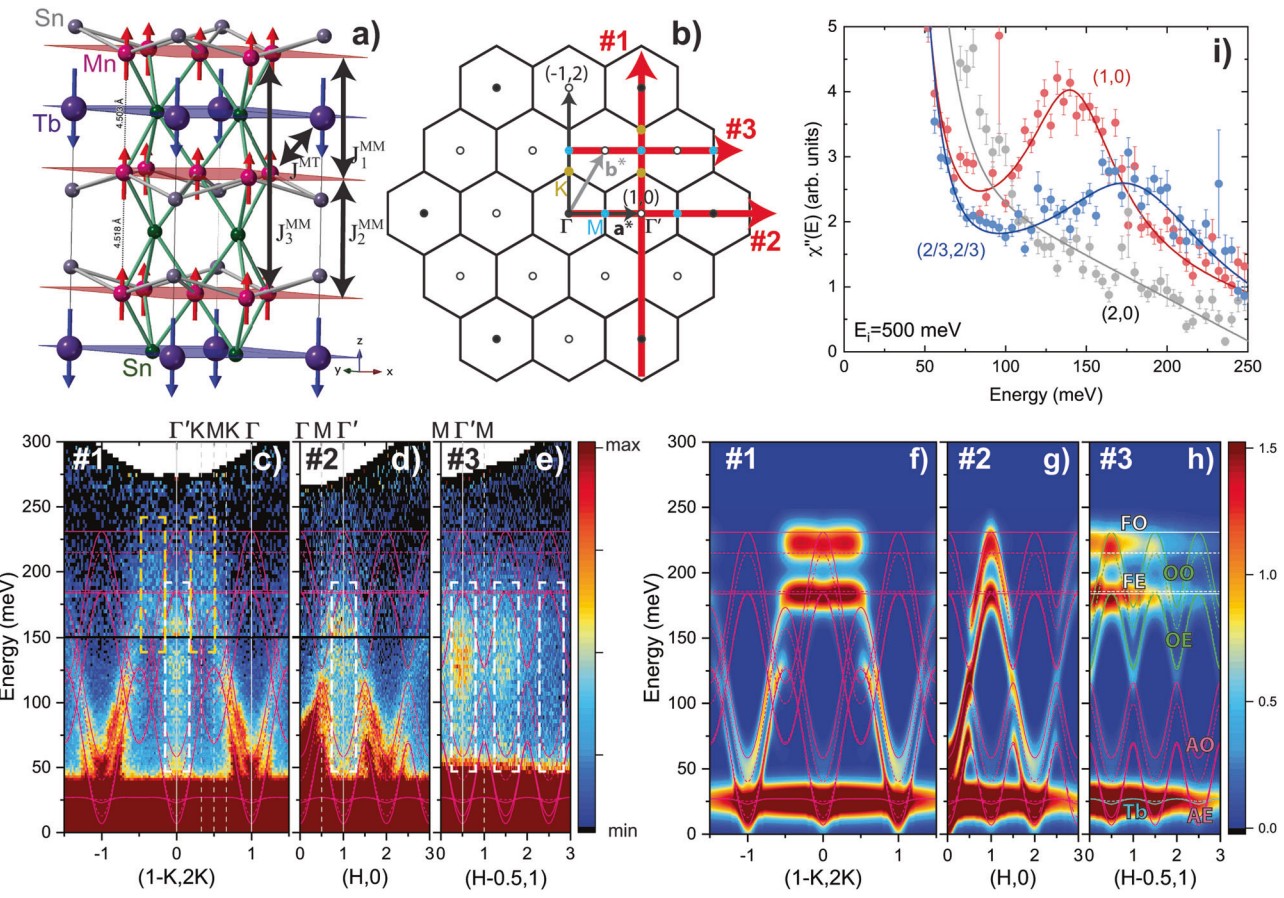

**Fig. 1 | Elementary magnetic excitations in TbMn$_6$Sn$_6$. a** Ferrimagnetic structure of TbMn$_6$Sn$_6$ with key interlayer magnetic interactions indicated by black arrows and where the superscripts M and T stand for Mn and Tb, respectively. **b** 2D hexagonal Brillouin zone showing reciprocal lattice vectors **a**$^*$ = (1, 0) and **b**$^*$ = (0, 1) and representative high symmetry Γ (black circles), Γ′ (empty circles), M (blue circles), and K (gold circles) points. INS data are described with the orthogonal vectors (1, 0) and (-1, 2). Red arrows correspond to the three reciprocal space slices shown in panels (**c–h**). Panels **c–e** show the intensity of slices through the INS data along the $(1 − K, 2K)$ (slice #1), $(H, 0)$ (slice #2), and $(H − 0.5, 1)$ (slice #3) directions, respectively, after averaging over $L = −7$ to 7 rlu. High-symmetry points are indicated by vertical dashed white lines. In panels **c** and **d**, data with $E < 150$ meV ($> 150$ meV) were collected with $E_i = 250$ meV (500 meV), respectively, as indicated by the black horizontal line. Data in (**e**) were collected with $E_i = 500$ meV. Panels **f–h** show

identical slices as **c–e** calculated from linear spin wave theory using the parameters described in the main text and averaged over $L = −7$ to -7 rlu. In panels **c–g**, the solid and dashed pink lines correspond to model dispersions with $L = 0$ and $L = 1/2$, respectively. In panel **h**, blue, pink, green, and white lines label the model dispersions of the Tb, AE/AO, OE/OO, and FE/FO modes, respectively, as described in the text. White (yellow) dashed rectangles outline overdamped excitations at the Γ′ (K) point, respectively. **i** Energy cuts centered at the Γ′ = (1, 0) point (red), at the K = $(\frac{2}{3}, \frac{2}{3})$ point (blue), and at the Γ = (2, 0) point (gray) with $E_i = 500$ meV. The data are scaled to the imaginary part of the dynamical susceptibility, $\chi''(E)$, after correcting for the $L$-averaged magnetic form factor. Error bars from counting statistics are one standard deviation. Solid lines correspond to damped harmonic oscillator fits, as described in the Supplementary Information.

optical–odd (OO), flat–even (FE), flat–odd (FO), and Tb. The even (odd) branches correspond to in-phase (out-of-phase) precession of Mn moments in adjacent kagome layers and have strong neutron intensity in Brillouin zones with $L = even$ ($L = odd$), respectively. The acoustic and optical branches are found below and above the K-point Dirac magnon crossing, respectively, and represent in-phase and out-of-phase precession of the three Mn moments within the unit cell of a single kagome layer. AE and AO branches have strong neutron intensity in Γ zones with $H = even$ and $K = even$, whereas OE, OO, FE, and FO branches are strongest in Γ′ zones with $H = odd$ and $K = odd$ or $H + K = odd$, as shown in Fig. 1b.

Previous INS experiments were conducted on smaller samples and mapped out the lowest-lying AE, AO, and Tb branches below 125 meV[22,23]. These magnons possess sharp, dispersive excitations throughout the Brillouin zone, indicative of their collective nature, and are well represented by a Heisenberg model consisting of intralayer and interlayer pairwise exchange interactions and single-ion anisotropy terms, as described in the Supplementary Information.

In ref. 22, we were unable to clearly observe the OE, OO, FE, and FO branches due to the small sample volume and increasingly broad line shapes encountered at higher energies. The measurements reported here reveal significant magnetic spectral weight up to 250 meV that accounts for the missing branches in the previous data. However, as previous reports hinted, these higher energy features are incoherent (broad in both momentum and energy), unlike the collective nature of the AO, AE, and Tb branches.

Figure 1 c–e show slices of the data along different reciprocal space directions within the kagome layers, as indicated in Fig. 1b. The data are averaged over $L$ to improve statistics, resulting in the simultaneous observation of even and odd branches, and are compared to the Heisenberg model dispersions shown as pink lines. We also compare the data to model calculations of the INS intensities with the same $L$-averaging, as shown in Fig. 1f–h.

In Fig. 1c and d, slices #1 and #2 along the $(1 − K, 2K)$ and $(H, 0)$ directions reveal dispersing AE and AO branches emanating from the Γ points which highlight their collective character. In slice #1, the AE and AO dispersions are well-defined up to their respective K-point Dirac

crossings at 90 meV and 140 meV. In slice #2, the AE and AO branches are well-defined up to the M-point with energies of 70 meV and 115 meV. The AE, AO, and Tb dispersions and intensities are consistent with model calculations shown in Fig. 1f–g.

Traces of broad excitations can be seen as high as ~250 meV in slices #1–3, consistent with model predictions of OE, OO, FE and FO modes with a 230 meV energy cutoff. However, the observed excitations are incoherent and heavily damped. Incoherent excitations observed in slices #1-3 form a steep feature centered in the Γ′ zones which extends from 180 meV down to surprisingly low energies of less than 50 meV (white rectangle). These modes are clearly observed in slice #3, where the Γ-zone AO and AE modes are suppressed. Slice #1 also indicates higher energy incoherent modes centered at the K-point and extending from 140–230 meV (yellow rectangle). Damped harmonic oscillator analysis of the energy spectra in Fig. 1i confirm lower average mode energies (145 and 190 meV) when compared to the Heisenberg model calculations. The peak widths are larger than both the experimental resolution (10-20 meV) and the broadening introduced from $L$-averaging of the interlayer bandwidth (20 meV). Combined with their damped character (with a quality-factor of 1.5–2), the heavily renormalized optical and flat modes resemble itinerant-like magnetic excitations[36–38].

In an attempt to capture some of these features, we investigate the extension of the current Heisenberg model to include longer-range interactions within the kagome layer. Additional FM interactions can be added with constraints that fix the M-point AO and AE energies while lowering the energy of the OO and OE branches at the Γ′-point. However, these interactions also introduce unsatisfactory distortions of the magnetic spectra (such as an overall lowering of the magnetic bandwidth). Ultimately, the broad nature of the high-energy excitations presents difficulties in the numerical fitting of the extended models. More details of these extended models are provided in Supplementary Fig. 1.

Rather than pursuing an extended Heisenberg model, a qualitative understanding of the incoherent high-energy excitations can be obtained from analyzing the momentum ($q$)-distribution of constant energy cuts through the excitation spectra, as shown in Fig. 2. Starting

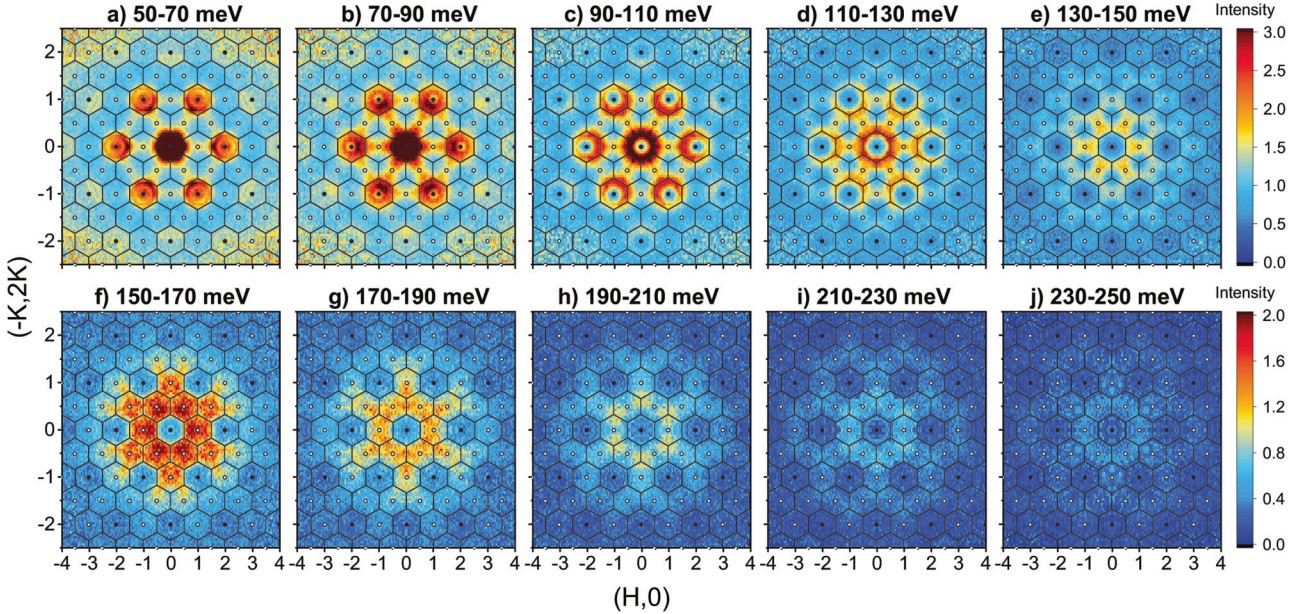

**Fig. 2 | Structure factor of magnetic excitations.** Constant energy slices of the TbMn$_6$Sn$_6$ spin excitations over incremental energy ranges from **a** 50–70 meV, **b** 70–90 meV, **c** 90–110 meV, **d** 110–130 meV, **e** 130–150 meV, **f** 150–170 meV, **g** 170–190 meV, **h** 190–210 meV, **i** 210–230 meV, **j** 230–250 meV. Data were collected at $E_i = 250$ meV for panels **a**–**e** and $E_i = 500$ meV for panels **f**–**j**. All cuts are averaged over an $L$-range from -7 to 7 rlu. Hexagonal Brillouin zone boundaries are shown and Γ and Γ′ zone centers are indicated by filled and empty circles, respectively.

at 60 meV in Fig. 2a, the intense AE and AO conical dispersions that form concentric rings of intensity in the Γ zones are observed. We also see ridges of intensity that extend longitudinally across the Γ′ zones. As energy is increased, the AO branch eventually reaches the zone boundary around 115 meV (Fig. 2d) while the longitudinal ridges in the Γ′ zone persist. At energies above the AO branch cutoff, only the ridge through the Γ′ zones remain (Fig. 2e). Increasing the energy further reveals that the Γ′ ridges overlap with scattering intensity that forms at K-points found at the intersection between Γ′ zones [eg., at $\mathbf{q} = \left(\frac{2}{3}, \frac{2}{3}\right)$]. Figure 2h shows that the K-point excitations are dominant at 200 meV. All magnetic excitations disappear at about 250 meV.

We make several conclusions about these results; (1) the expected OE/OO/FE/FO branches in Γ′ zones form two incoherent excitations, (2) one excitation forms longitudinal ridges across the Γ′ zones in Fig. 2c–e, and (3) the other excitation at the K-point extends to the highest energies as observed in Fig. 1c. For comparison, Heisenberg model calculations of these constant-E cuts can be found in Supplementary Fig. 2.

The broad momentum distribution of the high-energy modes suggests that the spin excitations are localized within the kagome layer. We studied both triangular (see Supplementary Fig. 4) and hexagonal plaquettes and conclude that the observed momentum distributions are accurately described by spin correlations that are localized on a single hexagon, as shown in Fig. 3a–c. The spin patterns are defined by the phase relationship between the instantaneous spin

components around the hexagon and shown for the case of transverse spin correlations with uniaxial magnetization along c.

The neutron intensity is estimated by calculating the corresponding static structure factors

$$S(\mathbf{q}) = f^2(q) \left| \sum_{j=1,6} e^{i\phi_j} e^{i\mathbf{q}\cdot\mathbf{r}_j} \right|^2 \tag{1}$$

as shown in Fig. 3d–f. Here, $f(q)$ is the magnetic form factor and $\phi_j$ describes the relative angle of the instantaneous spin direction of the transverse component of spin $j$. Dynamical spin precession averages over the spin direction at each site and only the relative angle between spins around the hexagon is relevant.

The FM spin correlations in Fig. 3a consist of in-phase precession of the spins ($\phi_{j+1} = \phi_j$), resulting in a momentum distribution with strong intensity in the Γ zones (Fig. 3d). This corresponds to the AO/AE modes, although these modes are more appropriately described as collective excitations using spin wave theory[22,23].

Figure 3c shows the expected flat-band spin correlations which consist of successive 180° rotations of spins around the hexagon ($\phi_{j+1} = \phi_j + \pi$). This is the correct Wannier representation of the localized flat-band excitations[28] and generates a momentum distribution which is peaked at the K-points surrounded by Γ′

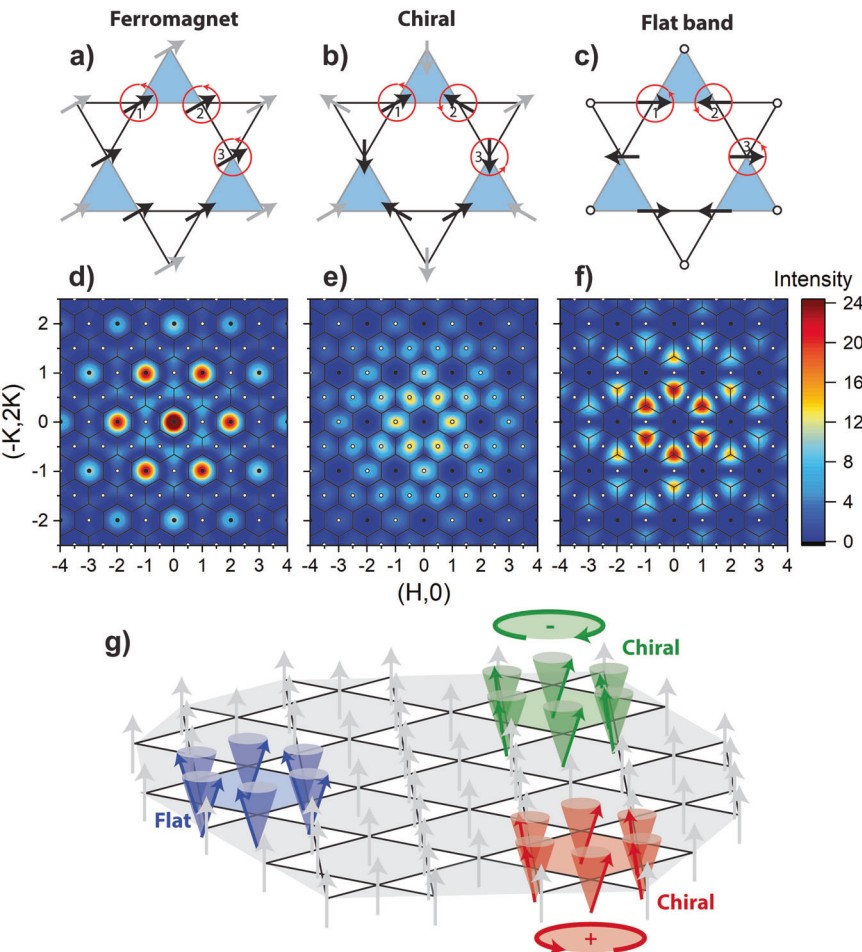

**Fig. 3 | Localized magnetic quasiparticles.** Spin patterns for an in-plane component of the Mn moment in **a** ferromagnetic, **b** chiral, and **c** flat-band spin correlations on a hexagonal plaquette. Numbers label the unique spins in the kagome unit cell and red circles indicate the direction of spin precession. Panels **d**–**f** show the corresponding static structure factors for the black spins around the hexagonal plaquettes in (**a**–**c**), respectively. **g** Localized flat band and chiral quasiparticles displayed as transverse excitations in a uniaxial FM kagome layer.

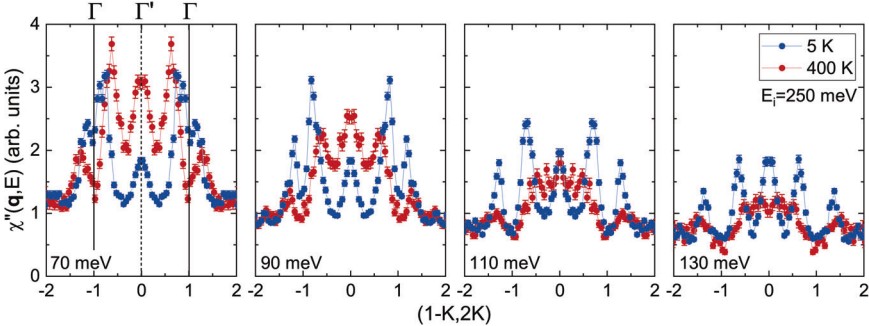

**Fig. 4 | Comparison of constant-energy cuts along the $(-K, 2K)$ direction for $T = 5\,K$ (blue) and 400 K (red).** Data are scaled to be proportional to the dynamical susceptibility, $\chi''(\mathbf{q},E) = I(\mathbf{q},E)(1 - \exp(-E/k_B T))$. Cuts are performed with $E_i = 250$ meV at energy transfers of 70, 90, 110, and 130 meV (averaged over an energy range of $\Delta E = \pm 5$ meV). All plots are averaged over reciprocal space ranges of $H = [0.9{:}1.1]$ and $L = [-7{:}7]$. Error bars from counting statistics are one standard deviation.

zones (Fig. 3 f). This momentum distribution is a signature of flat-band correlations[39].

The spin correlations in Fig. 3b are based on $q = 0$ AF order, which is the 120° chiral magnetic ground state found in kagome systems with nearest-neighbor AF interactions, such as iron jarosite[40,41]. In the long-range-ordered $q = 0$ AF state, magnetic Bragg peaks appear in $\Gamma'$ zones and are systematically absent in the $\Gamma$ zones. For chiral spin correlations, the Mn spins successively rotate by 120° around the hexagon ($\phi_{j+1} = \phi_j \pm 2\pi/3$) and one can define the vector chirality $\mathbf{n} = \frac{1}{3\sqrt{3}} \sum_{j=1,6} \mathbf{S}_j \times \mathbf{S}_{j+1}$. Figure 3e shows that chiral spin correlations have a momentum distribution consistent with longitudinal ridges observed in the $\Gamma'$ zones. Positive and negative chiralities have the same intensity pattern and cannot be differentiated. Thus, the two elementary excitations are associated with chiral and flat-band quasiparticles that are localized on an isolated hexagonal plaquette, as represented in Fig. 3g.

Knowledge of the direction (**n**) of the vector chirality can play a major role in understanding the origin of chiral quasiparticles. For example, if the chiral quasiparticles originate from transverse spin correlations about the magnetization direction (**M**), then $\mathbf{n} = \pm\hat{M}$ is (anti)parallel to the magnetization. A comparison of the excitations in the uniaxial and planar ferrimagnetic phases of TbMn$_6$Sn$_6$ at 5K and 400 K, respectively, demonstrates that chiral quasiparticles are present regardless of the magnetization direction. Figure 4 and Supplementary Fig. 3 show that both dispersive AO/AE modes in $\Gamma$ zones and localized $\Gamma'$ chiral modes are observed at 400 K. The intensity of both modes rapidly diminishes above 100 meV as a consequence of softening and mode damping near $T_C$, confirming that they originate from the Mn kagome layers. Calculations described in Supplementary Fig. 5 show that neutron scattering measures identical momentum space patterns for chiral spin correlations in the uniaxial and domain-averaged planar phases, indicating that unpolarized neutron scattering cannot determine the direction of **n**. Future polarized neutron scattering or other experimental data are necessary to resolve this important question.

## Discussion

The uniaxial FM kagome layers in TbMn$_6$Sn$_6$ host conventional, collective acoustic magnon quasiparticles representing the in-phase precession of Mn moments within a kagome layer. The thermal population of acoustic spin-1 magnons controls the magnetization of the kagome layer. However, chiral (optical) and flat-band modes whose precessions are out-of-phase have an incoherent character and remain localized to a single hexagonal plaquette with heavy damping.

The confinement of the flat magnon band to a hexagonal plaquette is known to arise from the special kagome lattice geometry which leads to phase cancellation of spin precession on the triangular

vertices surrounding a hexagon. The confinement of spin excitations to hexagonal plaquettes in frustrated pyrochlore AFs is similarly related to flat-band modes[42]. Thus, the flat bands observed here would, in principle, be consistent with linear spin wave theory were it not for the heavy damping. Similar to their electronic counterparts, the observation of a flat magnon band is rare and to our knowledge has never been unambiguously reported in metallic kagome magnets. Only in insulating kagome FMs, where damping is small, has such a flat band has been observed, e.g., Cu[1,3-benzenedicarboxylate][43]. In kagome AFs, localized flat-band magnon quasiparticles have been associated with novel magnetization processes that lead to magnon crystallization at high concentrations[44,45].

The observation of chiral magnetic quasiparticles is surprising and not expected from linear spin wave theory. Unlike the flat-band magnons, the kagome lattice geometry does not guarantee the localization of chiral excitations to a hexagon. Emergent chiral magnetic quasiparticles are often associated with topological, vortex-like spin textures, such as skyrmions, that are localized by a balance of exchange interactions and spin-orbit coupling[46]. Skyrmions have a non-zero scalar spin chirality where broken inversion symmetry of the crystal lattice imparts a handedness to the quasiparticle's spin texture. For TbMn$_6$Sn$_6$, spatial inversion is preserved, yet the observed chiral magnetic quasiparticles also consist of vortex-like excitations that possess a vector spin chirality with left and right-handed versions as shown in Fig. 3**g**. Such excitations can be associated with AF kagome systems which exhibit ground state configurations that carry a non-zero vector spin chirality. For example, elementary chiral excitations are observed above $T_N$ in $q = 0$ chiral AFs[41].

In TbMn$_6$Sn$_6$, the soft-mode character of the chiral excitations could indicate an instability of the FM kagome layers towards $q = 0$ AF order. Conditions for a soft-mode instability of chiral magnons may arise from frustrated pairwise interactions. A similar scenario has been proposed for some FM kagome metals such Co$_3$Sn$_2$S$_2$, where itinerant FM order is proposed to coexist with $q = 0$ chiral AF order[47–49]. While FM order is robust in TbMn$_6$Sn$_6$ ($T_C = 420$ K) and extended Heisenberg models are unlikely to generate chiral soft modes, some other mechanism, such as a band-driven magnetic instability, could be at play.

Along these lines, other clues to this anomalous behavior come from the strong damping that leads to short lifetimes of chiral and flat-band quasiparticles. Magnon damping can sometimes be associated with mode-mode coupling in local moment systems close to $T_C$ or even coupling to phonons. However, the strong damping observed at low temperatures and high energies (above the phonon cutoff) would be inconsistent with these mechanisms. Rather, the damping observed in TbMn$_6$Sn$_6$ and other FM kagome metals, such as YMn$_6$Sn$_6$[24] and FeSn[25–27], is more consistent with the Landau damping caused by the decay of magnons into particle-hole excitations (Stoner excitations).

Recent ab initio calculations for FeSn are consistent with heavy Landau damping for magnons above 80 meV[50]. Supplementary Fig. 7 shows DFT calculations of the bare electronic spin-flip susceptibility that support Landau damping above 50 meV for $TbMn_6Sn_6$. Electronic orbital currents, predicted to circulate on the same plaquettes[9,10], could impart a vector chirality perpendicular to the kagome layer and provide a mechanism for localization and damping of the chiral quasiparticles. Landau damping channels should also consider the conservation of chirality in the decay of chiral quasiparticles.

## Materials and methods
### Neutron scattering measurements
Single crystals of Tb166 were grown from excess Sn using the flux method as previously described[22]. INS measurements were performed on the Wide Angular-Range Chopper Spectrometer (ARCS) at the Spallation Neutron Source at Oak Ridge National Laboratory. An array of nine crystals with a total mass of 2.56 grams was co-aligned with the $(H,0,L)$ scattering plane set horizontally, and attached to the cold head of a closed-cycle-refrigerator. The data were collected at the base temperature of 5 K and 400 K using incident energies of $E_i$ = 250 and 500 meV. For each $E_i$ measurement, the sample was rotated around the vertical axis to increase the $\mathbf{q}$ coverage. The neutron scattering data are described using the momentum transfer in hexagonal reciprocal lattice units, $\mathbf{q}(H,K,L) = \frac{2\pi}{a}\frac{2}{\sqrt{3}}(H\hat{a}^* + K\hat{b}^*) + \frac{2\pi}{c}L\hat{z}$. The INS data are presented in terms of the orthogonal vectors $(1, 0, 0)$ and $(-1, 2, 0)$, as shown in Fig. 1**b**. We describe the data with reference to special points in the 2D Brillouin zone; $\Gamma-(0,0)$, $M-(\frac{1}{2},0)$, and $K-(\frac{1}{3},\frac{1}{3})$. The INS data are displayed as intensities that are proportional to the spin-spin correlation function $S(\mathbf{q}, E)$, where $E$ is the energy. To improve statistics, the data have been symmetrized with respect to the crystallographic space group P6/$mmm$.

## Data availability
Source data for line and scatter plots are provided in this paper. Inelastic neutron scattering data analyzed here can be obtained in the MDF open data repository[51,52] with the identifier https://doi.org/10.18126/FU9M-Y02F. Associated analysis and reduction scripts are available from R.J.M. upon reasonable request.

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

## Acknowledgements

RJM would like to thank Joe Checkelsky and Farhan Islam for comments. RJM, LK, PPO, BGU, BL, and SXMR's work at the Ames Laboratory is supported by the U.S. Department of Energy (USDOE), Office of Basic Energy Sciences, Division of Materials Sciences and Engineering. TJS, TH, and PC are supported by the Center for the Advancement of Topological Semimetals (CATS), an Energy Frontier Research Center funded by the USDOE Office of Science, Office of Basic Energy Sciences, through the Ames Laboratory. Ames Laboratory is operated for the USDOE by Iowa State University under Contract No. DE-AC02-07CH11358. A portion of this research used resources at the Spallation Neutron Source, which is a USDOE Office of Science User Facility operated by the Oak Ridge National Laboratory.

## Author contributions

R.J.M., B.G.U., S.X.M.R., B.L., T.H., and D.L.A. conducted and analyzed INS experiments. P.C.C. and T.J.S. grew and characterized single-crystals for INS measurements. L.K. and P.P.O. provided theoretical support.

## Competing interests

The authors declare no competing interests.
