## [Peer Review File · Nature Communications]

Chiral and flat-band magnetic quasiparticles in ferromagnetic and metallic kagome layersREVIEWER COMMENTS

Reviewer #1 (Remarks to the Author):

The paper titled "Chiral and flat-band magnetic quasiparticles in ferromagnetic and metallic Kagome layers" investigates the magnetic excitations of a Ferrimagnetic Kagome metal TbMn6Sn6 by using TOF inelastic neutron scattering. Except for previously identified sharp, collective acoustic magnon modes, the Authors observed two new elementary excitations with heavy damping at higher energies in larger sample volumes which one is identified as the flat-band magnon quasiparticles localized on an isolated hexagonal plaquette of Kagome layer and the other one is claimed to associated with the chiral magnetic quasiparticles. The INS experimental results were well shown and well analyzed by using a simple Heisenberg model including proper exchange interactions and single-ion anisotropy. The static structure factors from various in-plane spin components were calculated to qualitatively distinguishing between the magnetic quasiparticles. Besides, the damping effect of the flat-band and chiral excitations were further studied by temperature dependence and properly discussed with the help of DFT calculations. The authors conclude that the short-lifetime of the flat-band and chiral magnetic quasiparticles arise from a strong hybridization between the charge and the magnetic sector.

Obviously, the neutron experiments were beautifully done, the data quality is very good, the models used were very simple and easy to understand, the writing of descriptions in this paper are also good, and the explanations of the two anomaly magnetic excitations were quite convincing. The author's work is quite important for understanding the magnetic interactions, new exotic quasiparticles and interplay between charge and magnetism in such kind of Kagome magnetic metals.

However, before I recommend this work to publish on NC, I have some minor questions and comments. Please find the issues I have raised in the following.

[1] I noticed that there is a spin reorientation transition occurs in TbMn6Sn6 at $T \sim 320\text{K}$ and the easy magnetization direction will change from c-axis at low temperatures to the ab-plane at high temperatures (COMMUNICATIONS PHYSICS (2022) 5:107 <https://doi.org/10.1038/s42005-022-00885-4>). Principally, the spin reorientation will have an effect on chiral and flat-band excitations, but it seems the section of the main text on temperature dependence (5K and 400K) does not mention this effect.

[2] To improve the statistics, the neutron data have been averaged over $L=0-7$ in this paper, the dispersion of spin waves along (00L) direction (although not shown) may also play an important role on the broadening of the averaged results, especially for the optical magnon branches, how much broadening would such a large average result in? and how much would it affect or contribute to the damping effect, since optical branches are around similar energy range with chiral and flat-band excitations?

[3] Landau damping is one of several mechanisms that can contribute to the damping of spin waves in magnetic materials, the exclusion of other possibilities has not been discussed much. Maybe a simple comparison would highlight the plausibility of Landau damping in RMn6Sn6 system here. (Just a comment)

[4] Identify the continuum excitation at Γ in BZ as the localized chiral quasiparticles, sometimes spin fluctuations/magnetic disorder can give rise to a continuum excitation when the materials approach its instability/critical point, is there any other possible explanation for the anomalous excitation? Page 5, last paragraph: "In TbMn6Sn6, the soft-mode character of the chiral excitations could indicate an instability of the FM Kagome layers towards $q = 0$ AF order." Is that means TbMn6Sn6 can be driven into a pure chiral magnetic excitation (or chiral magnetic excitation dominated) phase with a proper applied field and temperature?

[5] In the calculation of spin wave, how to determine the magnetic moment size of Tb and Mn in such itinerant magnetic system, I noticed the spin wave theory calculation directly use $S=3$ and $S=1$ for localized moments of Tb and Mn respectively. I also noticed that the interaction exchange parameters used in this paper is not exactly the same as that in the author's previous work (PHYSICAL REVIEW X 12, 021043 (2022)), such as J^{MT} and the sign of Mn anisotropy term K^{M} .

[6] Page4 in the main text, the 1st sentence of the description of Fig.4. "Figure 4 and Supplementary Fig.4 compare cuts..." I suppose it is "Supplementary Fig. S3", right?

Reviewer #2 (Remarks to the Author):

Quantum materials that exhibit flat electronic and/or bosonic bands (excitations) became a research forefront in the condensed matter and material physics community in the past few years. Premium examples are bilayer graphene which shows unconventional superconductivity and kagome materials which display exotic topological, magnetic, and superconducting properties.

In the last few years, there are a number of experimental reports of observation of flat electronic bands in Kagome materials. Nearly flat magnetic excitations (magnetic band) was observed recently in $\text{Co}_3\text{Sn}_2\text{S}_2$. [Nature Communications 13, 7317 (2022)] In the meantime, the special geometry of the Kagome lattice can also give rise to a flat magnetic band (optical magnon) in the localized limit such as the Heisenberg model used in this manuscript. Nevertheless, there are very few examples of Kagome materials with flat magnetic band.

In this manuscript, the authors used inelastic neutron scattering to measure the magnetic excitations in a ferromagnetic kagome metal TbMn_6Sn_6 . The author claimed to observe sharp collective acoustic magnons and flat-band magnons.

The central piece of evidence to support the authors' claim is shown in Fig. 1 (c, d, e). According to the linear spin wave theory calculations shown in Fig. 1 (f, g, h), there are flat magnetic excitations near Gamma point at about 180 meV and 225 meV. However, the resolution of the experimental data shown in Fig. 1 (c, d, e) is too poor to identify the flat magnetic excitations around Gamma point. Other features shown in Fig. 1 (f, g, h) cannot be clearly identified in Fig.1 (c, d, e) either. In my opinion, the presented experimental data cannot support the claims made by the authors. Therefore, I am unable to support to publish the paper in Nature Communications.

The paper titled “Chiral and flat-band magnetic quasiparticles in ferromagnetic and metallic Kagome layers” investigates the magnetic excitations of a Ferrimagnetic Kagome metal TbMn_6Sn_6 by using TOF inelastic neutron scattering. Except for previously identified sharp, collective acoustic magnon modes, the Authors observed two new elementary excitations with heavy damping at higher energies in larger sample volumes which one is identified as the flat-band magnon quasiparticles localized on an isolated hexagonal plaquette of Kagome layer and the other one is claimed to associated with the chiral magnetic quasiparticles. The INS experimental results were well shown and well analyzed by using a simple Heisenberg model including proper exchange interactions and single-ion anisotropy. The static structure factors from various in-plane spin components were calculated to qualitatively distinguishing between the magnetic quasiparticles. Besides, the damping effect of the flat-band and chiral excitations were further studied by temperature dependence and properly discussed with the help of DFT calculations. The authors conclude that the short-lifetime of the flat-band and chiral magnetic quasiparticles arise from a strong hybridization between the charge and the magnetic sector. Obviously, the neutron experiments were beautifully done, the data quality is very good, the models used were very simple and easy to understand, the writing of descriptions in this paper are also good, and the explanations of the two anomaly magnetic excitations were quite convincing. The author’s work is quite important for understanding the magnetic interactions, new exotic quasiparticles and interplay between charge and magnetism in such kind of Kagome magnetic metals.

We thank the Referee for their very supportive comments.

However, before I recommend this work to publish on NC, I have some minor questions and comments. Please find the issues I have raised in the following. [1] I noticed that there is a spin reorientation transition occurs in TbMn_6Sn_6 at $T \sim 320\text{K}$ and the easy magnetization direction will change from c-axis at low temperatures to the ab-plane at high temperatures (COMMUNICATIONS PHYSICS (2022) 5:107 <https://doi.org/10.1038/s42005-022-00885-4>). Principally, the spin reorientation will have an effect on chiral and flat-band excitations, but it seems the section of the main text on temperature dependence (5K and 400K) does not mention this effect.

The Referee makes an excellent point regarding the effect of the spin reorientation transition on the chiral and flat-band excitations. The data in Figure 4 indicate that the chiral excitations are present regardless of the direction of the magnetization. Unfortunately, we do not have corresponding data for the flat-band excitations at 400 K, so we focus mainly on the chiral excitation in the discussion below.

In the original manuscript, Fig. 3 shows calculations of the momentum-space distribution for chiral and flat-band spin correlations assuming that they are transverse to the uniaxial magnetization direction pointing along z. In that case, the distinctive patterns in the neutron intensity are caused by spin correlations between the x and y spin components with vector chirality (anti)parallel to z. We have now done more detailed calculations of the momentum distribution for transverse spin correlations in the planar phase where the magnetization lies in the xy-plane (for example, we calculate the spin correlations of the y and z components if the moment points along the x-direction). The conclusion is that the momentum-space patterns in the planar and uniaxial phases are identical (after averaging over the possible domains in the planar phase). Results from

these calculations are shown in Supplementary Fig. 5 and their description have been added to a new section in the Supplementary Information titled “Chiral spin correlations with planar magnetization”.

Thus, the direction of the vector chirality cannot be determined from our zero-field INS data and the chirality may or may not be (anti)parallel to the Mn layer magnetization. This can have consequences for the physical interpretation chiral excitations. For example, if the vector chirality remains (anti)parallel to the magnetization, then we may associate it with transverse spin correlations of the Mn magnetization. On the other hand, if the vector chirality is unchanged across the spin reorientation, it could indicate a more exotic origin such as orbital loop currents where the vector chirality is perpendicular to the Mn layer. Future experiments, such as polarized neutron scattering, may be able to resolve this issue and obtain a better understanding of the chiral excitations. Based on this new information, we have updated the discussion points in the main manuscript regarding the vector chirality.

[2] To improve the statistics, the neutron data have been averaged over $L=0-7$ in this paper, the dispersion of spin waves along (00L) direction (although not shown) may also play an important role on the broadening of the averaged results, especially for the optical magnon branches, how much broadening would such a large average result in? and how much would it affect or contribute to the damping effect, since optical branches are around similar energy range with chiral and flat-band excitations?

Fig. 1 panels (f)-(h) (also in Fig. S2 in the supplement) show model calculations that have been averaged over L in the same manner as the experimental data. These calculations make it clear that averaging over the L -dependent dispersions can broaden certain excitations, such as acoustic odd modes and flat-band odd modes, by as much as 20 meV. This can be seen most clearly in the experimental data as broadening of the acoustic odd modes [Figs. 1 (c) and (d)] which indicates that our models have the correct L -dependent bandwidths (See also Riberolles, et al., Phys. Rev X 12, 021043 (2022), and Riberolles, et al., Nat. Comm. 14, 2658 (2023)). The observed broadening of the flat and chiral modes is substantially larger (69 and 115 meV) than 20 meV and therefore must have an origin other than L -averaging. We have added text to the Fig. 1 caption indicating that model calculations are L -averaged and the main manuscript on p. 2 now clearly states that L -averaging cannot account for the observed broadening.

[3] Landau damping is one of several mechanisms that can contribute to the damping of spin waves in magnetic materials, the exclusion of other possibilities has not been discussed much. Maybe a simple comparison would highlight the plausibility of Landau damping in RMn_6Sn_6 system here. (Just a comment)

Other possibilities include mode-mode coupling and magnetoelastic interactions (i.e. coupling to phonons) which is now introduced in the Discussion on p.6. For the former, it is expected that mode-mode coupling becomes significant when the moment fluctuations become large and outside the linear spin wave theory approximation. For local moment systems, this is typically the case when the temperature is close to T_C and thermal fluctuations are large. However, this scenario is unlikely for TbMn_6Sn_6 where measurements are taken well below $T_C = 420$ K. Also, as seen in the data, the energy of the chiral and flat band modes is much larger than the phonon bandwidth, making coupling to phonons an unlikely origin for the large damping.

[4] Identify the continuum excitation at Γ in BZ as the localized chiral quasiparticles, sometimes spin fluctuations/magnetic disorder can give rise to a continuum excitation when the materials approach its instability/critical point, is there any other possible explanation for the anomalous excitation? Page 5, last paragraph: “In TbMn_6Sn_6 , the soft-mode character of the chiral excitations could indicate an instability of the FM Kagome layers towards $q = 0$ AF order.” Is that means TbMn_6Sn_6 can be driven into a pure chiral magnetic excitation (or chiral magnetic excitation dominated) phase with a proper applied field and temperature?

Indeed, the sentence highlighted by the Referee shows that we consider this to be a possibility. The chiral mode does resemble an overdamped soft mode that is perhaps driven by frustrated magnetic interactions within a kagome layer. We did consider some extended Heisenberg models in the Supplementary Information (Fig. S1) which consider exactly this point, but there was no simple or obvious solution that generated a soft mode without introducing strong disagreement with other parts of the spectrum. There may be other explanations for the origin of a soft-mode, such as band-driven magnetic interactions or hybridization with carriers. This is now mentioned explicitly in the Discussion on p. 6. Whether there is a mechanism whereby the soft mode can be driven to $q=0$ AF order, perhaps by chemical substitution, is a very interesting and exciting possibility that we intend to pursue in future work.

[5] In the calculation of spin wave, how to determine the magnetic moment size of Tb and Mn in such itinerant magnetic system, I noticed the spin wave theory calculation directly use $S=3$ and $S=1$ for localized moments of Tb and Mn respectively. I also noticed that the interaction exchange parameters used in this paper is not exactly the same as that in the author’s previous work (PHYSICAL REVIEW X 12, 021043 (2022)), such as J^{MT} and the sign of Mn anisotropy term K^{M} .

This discrepancy has to do with the handling of the single-ion anisotropy in linear spin wave theory codes. Typically, the exchange coupling occurs through the spin S and the CEF anisotropy is represented with operators of the total angular momentum J (Stevens operators). This leads to problems in spin wave software packages, such as SpinW, where each ion can only have a single-valued spin quantum number. In the PRX, the data was fit using linear spin wave theory with all terms containing the spin of the Tb ion ($S=3$). This choice affects the calculated spin gap of the Tb mode and leads to issues when fitting the Tb and Mn single-ion anisotropies, as noted in the PRX (first column, page 6 of that manuscript). We subsequently analyzed the spin excitations within the random-phase approximation (RPA), where terms in the magnetic Hamiltonian can be properly expressed in terms of both S and J operators [see Riberolles et al., Nat. Comm. 14 2658 (2023)]. From this analysis, we realized, perhaps counterintuitively, that the correct approach within linear spin wave theory is to assign a total angular momentum $J=6$ to the Tb ion. Refitting of the same data in the PRX using linear spin wave theory package SpinW with $J=6$ rather than $S=3$ leads to updated values for the J^{MT} , K^{M} , and K^{Tb} as reported in the Nat. Comm article and utilized in the RPA approach. It does not change the J^{MM} interactions. This subtle and technical point about scaling of interactions in complex magnetic Hamiltonians is discussed in more detail in Supplementary Note I of the Nat. Comm. 14 2658 (2023), article and we now reference that article in both the main manuscript and the Supplement.

[6] Page4 in the main text, the 1st sentence of the description of Fig.4. “Figure 4 and Supplementary Fig.4 compare cuts...” I suppose it is “Supplementary Fig. S3”, right?

You are correct and this typo has been fixed.

Reviewer #2 (Remarks to the Author):

Quantum materials that exhibit flat electronic and/or bosonic bands (excitations) became a research forefront in the condensed matter and material physics community in the past few years. Premium examples are bilayer graphene which shows unconventional superconductivity and kagome materials which display exotic topological, magnetic, and superconducting properties.

In the last few years, there are a number of experimental reports of observation of flat electronic bands in Kagome materials. Nearly flat magnetic excitations (magnetic band) was observed recently in $\text{Co}_3\text{Sn}_2\text{S}_2$. [Nature Communications 13, 7317 (2022)] In the meantime, the special geometry of the Kagome lattice can also give rise to a flat magnetic band (optical magnon) in the localized limit such as the Heisenberg model used in this manuscript. Nevertheless, there are very few examples of Kagome materials with flat magnetic band.

In this manuscript, the authors used inelastic neutron scattering to measure the magnetic excitations in a ferromagnetic kagome metal TbMn_6Sn_6 . The author claimed to observe sharp collective acoustic magnons and flat-band magnons.

The central piece of evidence to support the authors' claim is shown in Fig. 1 (c, d, e). According to the linear spin wave theory calculations shown in Fig. 1 (f, g, h), there are flat magnetic excitations near Gamma point at about 180 meV and 225 meV. However, the resolution of the experimental data shown in Fig. 1 (c, d, e) is too poor to identify the flat magnetic excitations around Gamma point. Other features shown in Fig. 1 (f, g, h) cannot be clearly identified in Fig.1 (c, d, e) either. In my opinion, the presented experimental data cannot support the claims made by the authors. Therefore, I am unable to support to publish the paper in Nature Communications.

The resolution of the experiment is not a limitation. Rather the optical and flat band modes are overdamped and have peak widths that are many times larger than the experimental resolution. We have added the following sentence on p. 2 of the resubmitted manuscript: “The peak widths are larger than both the experimental resolution (10-20 meV) and the broadening introduced from L -averaging of the interlayer bandwidth (~ 20 meV).” We argue that this damping is intrinsic and likely due to interaction with charge carriers within the kagome layers. Despite their overdamped character, we can identify contributions of a flat band and, unexpectedly, a chiral band through careful analysis of the momentum distributions at different energies. Not only does this provide conclusive evidence for these novel excitations, but the damping itself is an important attribute that highlights a strongly itinerant character to these modes. We anticipate that these

clarifications referencing the quality of our data and the intrinsic character of mode damping will encourage the referee to reconsider their position on our manuscript.

REVIEWERS' COMMENTS

Reviewer #1 (Remarks to the Author):

I appreciate the effort the authors have put into addressing all the raised questions, and I find your explanations and revisions to be satisfactory.

One minor point, another typo in Page 6 of the main text from your revised version. Sentence "While FM order is robust in TbMn₆Mn₆ (TC = 420 K) ...", check TbMn₆Sn₆.

Once this is corrected, I believe the manuscript will be ready for publication.

Final response to referee.

We would like to thank the referee for pointing out a typo and confirm that we have corrected it on the submitted version.